# Effect of Heat Input on the Ballistic Performance of Armor Steel Weldments

**DOI:** 10.3390/ma14133617

**Published:** 2021-06-29

**Authors:** Branko Savic, Aleksandar Cabrilo

**Affiliations:** 1The Higher Education Technical School of Professional Studies in Novi Sad, Skolska 1, 21000 Novi Sad, Serbia; 2Faculty of Technical Sciences, University of Novi Sad, Trg D. Obradovića 6, 21000 Novi Sad, Serbia; cabrilo@uns.ac.rs

**Keywords:** armor steel, weldment, thermal history, projectile penetration, hardness level

## Abstract

The purpose of this study is to examine the projectile penetration resistance of the base metal and heat-affected zones of armor steel weldments. To ensure the proper quality of armor steel welded joints and associated ballistic protection, it is important to find the optimum heat input for armor steel welding. A total of two armor steel weldments made at heat inputs of 1.29 kJ/mm and 1.55 kJ/mm were tested for ballistic protection performance. The GMAW welding carried out employing a robot-controlled process. Owing to a higher ballistic limit, the heat-affected zone (HAZ) of the 1.29 kJ/mm weldment was found to be more resistant to projectile penetration than that of the 1.55 kJ/mm weldment. The ballistic performance of the weldments was determined by analyzing the microstructure of weldment heat-affected zones, the hardness gradients across the weldments and the thermal history of the welding heat inputs considered. The result showed that the ballistic resistance of heat affected zone exist as the heat input was decreased on 1.29 kJ/mm. It was found that 1.55 kJ/mm does not have ballistic resistance.

## 1. Introduction

Armor grade steels possessing high strength and hardness are widely used in the production of military armored vehicles such as Lazar III [1,2,3]. High-hardness armor steel requires carefully controlled welding procedures to avoid hardness losses in heat-affected zones [4,5,6]. Heat input is the crucial factor associated with the toughness of fusion zones in shielded-metal-arc-welding weldments [7,8,9]. The hardness of armor steel is greatly dependent on the welding temperature history. The heat-affected zone softening is effectively controlled by maintaining a high peak temperature gradient close to the weld bead [10]. The microstructure and width of the heat-affected zone (HAZ) is a function of the cooling rate imposed by the welding process and the CCT diagram of the base metal plate [11]. The microstructure of the HAZ affects the hardness level and the ballistic performance of weld joints. Heat input control is aimed at limiting the HAZ width within 15.9 mm, measured from the weld centerline, which is compliant with the standard MIL-STAN-1185 [12].

HAZ softening occurring during welding of HHA steel and the degree of softening in the HAZ is a function of the weld thermal cycle, which depends on the welding process [13]. In the welding process of military vehicles, two processes are predominantly used: metal inert gas welding (GMAW) and electric arc welding (MMA). GMAW process has a higher deposition rate, compared to the Shield metal arc welding [14]. In application of the GMAW process, the consumable is continuously added and frequent stops are not happening. As a result, the GMAW process has superior productivity compared to SMAW [15]. The pulsed GMAW process can be used in welding armor steel [16] and yields higher productivity than the conventional GMAW process.

The combination of GMAW and laser processes can achieve higher productivity than conventional GMAW [17]. However, the GMAW process results in higher ductility of welded joints, than hybrid laser GMAW process [18]. For multi-purpose military transport vehicles, which often negotiate rough terrains, low ductility is unacceptable.

Dissolved hydrogen and the hardness are essential parameters of cold cracking tendency in the fusion line and coarse-grained heat affected zone. Higher amount of hydrogen and higher volume content of hard martensite structures present in the HAZ near the fusion boundary result in higher susceptibility for cold cracking. It is necessary to choose appropriate welding parameters to achieve the most beneficial compromise between hardness and toughness.

Previous studies have shown that a heat input of 1.2 kJ/mm is safe for the ballistic protection of military armored vehicles, whereas a heat input of 1.9 kJ/mm has been found ballistically unsafe for the armor protection of military vehicles. However, a heat input of 1.2 kJ/mm was found herein to be insufficient to ensure the proper quality of armor steel welded joints, so it was of paramount importance to find the optimum heat input for obtaining the best ballistic protection of the welded joints. Appropriate welding parameters are essential for the ballistic resistance of weld joints [19,20,21] in military vehicles, as well as their toughness when the vehicles are moving over uneven terrains.

The CCT cooling rate diagram of armor steels provides important information for selecting the proper heat input of the welding process. When armor steel is more quenchable, i.e., pure martensite is obtained at a slow cooling rate of <12 °C/s, a higher heat input is allowed, compared to armor steels where a higher cooling rate of <25 °C/s is required to achieve pure martensite. According to this statements, Protac 500 require increase preheat and inter pass temperatures.

Precise trajectories of the robotic arm movement are necessary for heat input testing. Industrial robots have wide applications due to their ability to perform operations quickly, repeatedly and accurately. Process flexibility and the possibility of welding in various positions are the main reasons for using robots [22]. Therefore, the run trajectory and the position of a gun relative to the weld axis must be maintained accurately.

This paper presents a comparison of the ballistic performance of quenchable armor steel weldments made at heat inputs of 1.29 kJ/mm and 1.55 kJ/mm, which form a 100% martensitic structure at a cooling rate of 7 °C/s.

## 2. Materials and Experiment

### 2.1. Base and Filler Material

Armor steels are well established as projectile-resistant materials. The commercially available Protac 500 armor steel was used in this study for its high strength (σy = 1206 MPa and UTS = 1536 MPa) [23].

The chemical composition of base metal and weld metal was carried out by optical emission spectrometer ARL 2460 Metals Analyzer, (ThermoFisher, Waltham, MA, USA) and results are given in Table 1.

### 2.2. Welding Process

Gas metal arc welding (GMAW) was performed at heat inputs of 1.29 kJ/mm and 1.55 kJ/mm, using the same welding configuration (Figure 1). A 55-degree single V groove edge with a 4-mm root face gap was employed before welding. Each weld was produced by four-pass welding with pre-heating. The plate dimensions were 500 mm × 250 mm × 11 mm. A water jet cutter was employed for plate cutting and edge preparing.

The Protac 500 welding parameters are shown in Table 2. Welding heat input calculated in accordance with EN 1011-1, using equitation 1. Where heat transfer efficiency was 0.8 for GMAW.
(1)$$Heat\;input=\frac{Arc\;voltage*Arc\;current*heat\;tranfer\;efficiency}{Travel\;speed}$$

The welding process was performed by pushing (with a gun angle of 20°). The GMAW welding was performed in a shielding gas of argon +2.5% CO_2_ with flow rates of 10 L/min times the wire diameter and 12 L/min times the wire diameter. The wire diameter was 1.0 mm.

The automated welding for both heat inputs considered was performed using the Kuka robot, and the Citronix 400A GMAW welding machine (Augsburg, Germany).

### 2.3. Cooling Rate and Hardness Measuring

The first phase of testing involved measuring the cooling rate in the critical coarse-grained heat-affected zone (HAZ) of the welded (Protac 500) joints. The cooling rate was measured for each heat input considered to determine changes in temperature with time. To achieve uniform heating, the plates were preheated in a furnace to a few degrees above 150 °C. The NiCr-Ni thermocouples were used to measure the preheating temperature of the plates and the cooling rate of the welded joint HAZ. The position of thermocouple is shown in Figure 2. The temperature changes for heat inputs of 1.29 kJ/mm and 1.55 kJ/mm were measured with the thermocouple 1, placed 4 mm from the end and 1 mm from the lower plate edge. Temperature and time data were acquired through thermocouples connected to a data acquisition card and a computer. To validate the data acquired, a contact thermometer was also used as the secondary temperature measurement method.

The cooling time, t8/5, from 800 to 500 °C in weldment is usually selected to define the critical cycle. However, the continuous cooling transformation of the armor steel showed that microstructure transformation occurs in the temperature region of 600 to 200 °C. Due to this, t6/2, a cooling time from 600–200 more precisely represents cooling conditions crucial for evolution of microstructure [15].

According to the EN ISO 9015-1 standard [24], the hardness of welded joints is measured for their complete characterization. To reveal the microstructure samples were etched using 2% nital. Armor steel welded joints must meet the requirements stipulated by the MIL-STAN-1185 standard [12]. The hardness of the Protac 500 welded joints was herein tested 2 mm under the upper welding surface at heat inputs of 1.29 kJ/mm and 1.55 kJ/mm. The hardness of both heat input samples considered was measured along the fusion line for achieving optimum hardness in this critical zone and along the edge of the weld metal. The Digital Micro Vickers Hardness Tester HVS1000 (Laiznou Huayin Testing Instrument Co., Laizhou, China) was used for microhardness testing, applying a load of 500 g. Each microhardness value represents the mean value of three measurements performed.

### 2.4. Ballistic Testing

The ballistic resistance testing in this study was accomplished in accordance with the VPAM APR 2007 standard [25], which stipulates placing the ballistic pipe at a distance of 10 m from the target [26]. The ballistic test scheme and the 7.62 × 51 mm projectile used are shown in Figure 3a,b. The target was positioned on a backrest, at right angles to the level of shooting. The projectile speed was measured prior to the experiment at a distance of 7 m from the position of the ballistic pipe mouth. The projectile speed measurements were performed on three projectiles to obtain a representative mean value of the projectile speed.

## 3. Results and Discussion

This section may be divided by subheadings. It should provide a concise and precise description of the experimental results, their interpretation, as well as the experimental conclusions that can be drawn.

### 3.1. Instrumented Charpy Impact Test

Table 3 presents the results of the Charpy impact energy tests for the investigated samples in base metal. Each Charpy impact energy value represents the mean of the three measurements. The impact energy on the base metal was 35.2 J and 27.7 J at temperatures of 20 °C and −40 °C, respectively.

SEM fractographs of specimens fractured at 20 °C are shown in Figure 4a,b. The diagram presented in Figure 4a is characteristic for the ductile–brittle fracture. Higher energy (30.4 J) was absorbed on the crack initiation, while the crack growth absorbed significantly less (4.8 J). The impact energy diagram at −40 °C (Figure 4c), shows brittle fracture (Figure 4d). The measured impact energy for crack initiation in this zone was 24.9 J, with 2.9 J absorbed on the crack growth.

### 3.2. Cooling Rate

During the welding process, the weld thermal history was recorded for both heat inputs considered with the selected inter-pass temperature.

Figure 5 shows the weld thermal history of 4 passes deposited with heat inputs of 1.29 kJ/mm and 1.55 kJ/mm. The cooling rate of the 1.55 kJ/mm weldment was found to decrease with each pass, in contrast to the simultaneously increasing cooling time. The HAZ weld thermal history was recorded to provide information about the effects of the 1.29 kJ/mm and 1.55 kJ/mm heat inputs on the cooling rate and cooling time of the welded joint heat-affected zones.

As shown in Table 4, an increase in heat input from 1.29 kJ/mm to 1.55 kJ/mm increased the overall welding time. The cooling time measured from the peak temperature of the first cap pass to the peak temperature of the final pass was 1293 for the 1.55 kJ/mm weldment, compared to 1122 s recorded for the 1.29 kJ/mm weldment. An increase in heat input from 1.29 kJ/mm to 1.55 kJ/mm resulted in a cooling time decrease of approximately 2.9 min. The overall cooling time (from the peak temperature of the first cap pass (tp1st pass) to a temperature of 160 °C recorded upon completion of the welding process (t100 °C final pass)) was gradually reduced from 1696 s to 1484 s with an increase in heat input from 1.29 kJ/mm to 1.55 kJ/mm. This can be accounted for by the significantly longer cooling time after the deposition of the final weld pass. The overall cooling time data is important for assessing the time required for cooling relative to the diffusion of hydrogen out of the weldment during the welding process.

The temperature diagrams of changes in the cooling time of the 1.29 kJ/mm and 1.55 kJ/mm weldments are shown in Figure 5. As the data obtained exhibited a wide dispersion pattern, the regression line was fitted through 10 data points.

As shown in Table 4, the cooling time (Δt6/2) of the four-pass 1.29 kJ/mm armor steel welding was longer than that of the 1.55 kJ/mm welding. This can be attributed to a slightly lower welding speed and the impossibility of continuous gun movement.

### 3.3. Hardness

Hardness is one of the most important aspects of armored vehicle crew protection and the quality of welded joints. The hardness profiles obtained for heat inputs of 1.29 kJ/mm and 1.55 kJ/mm are shown in Figure 6. The hardness measurement results in Table 5 represent the mean value of three measurements performed.

Figure 6a shows the hardness of the welded joints preheated at 150 °C, using an inter-pass temperature of 160 °C. The welded joint zones were marked accordingly with the following abbreviations: WM (weld metal), FL (fusion line), HAZ (heat-affected zone), IZ (inter-critical zone), SZ (subcritical zone) and BM (base metal).

The hardness profile obtained for a heat input of 1.29 kJ/mm (Figure 6a) indicates hardness variations in the WM, FL, HAZ and BM zones. The hardness values increased from the middle of the WM zone (190 HV) towards the fusion line, along which a value of 339 HV was recorded on the WM side. The FL hardness value was 410 HV. The hardness values increased in the HAZ zone and reached a maximum value of 521 HV at a distance of 8 mm from the weld axis. The values decreased thereafter and a minimum hardness of 378 HV was recorded at a distance of 10 mm from the weld axis. Upon another subsequent increase, the hardness values eventually leveled off at 509 HV recorded at a distance of 14 mm from the weld axis, which also marked the limit of the HAZ and OM zones. The average BM hardness value was 509 HV.

The hardness profile obtained for a heat input of 1.55 kJ/mm (Figure 6b) also suggests hardness variations in the WM, FL, HAZ and BM zones. The hardness values increased from the middle of the WM (192 HV) towards the fusion line, along which a value of 350 HV was recorded on the WM side. The FL hardness was 400 HV. The hardness values decreased in the HAZ zone and reached a minimum value of 325 HV at a distance of 10.5 mm from the seam axis. Upon another subsequent increase, the hardness values eventually leveled off at 509 HV recorded at a distance of 14 mm from the seam axis, which also marked the limit of the HAZ and OM zones. The average BM hardness value was 509 HV.

The hardness of the 1.29 kJ/mm and 1.55 kJ/mm weldments was measured along the fusion line (Figure 6b). The results obtained show that the fusion line hardness of the 1.29 kJ/mm weldment ranged between 408 HV and 431 HV, whereas the fusion line hardness of the 1.55 kJ/mm weldment ranged between 398 HV and 421 HV. The hardness values were found to be associated with heat effects: the heat effect was more significant in the zones closer to the cover pass, whereas the already cooled additional and base material reduced the heat effect in the more remote zones.

### 3.4. Microstructure

Microstructure testing was carried out on the 1.29 kJ/mm and 1.55 kJ/mm weldments. The metal microstructures of both weldments considered (Figure 7a,b) consisted of the austenitic base and δ-ferrite. The following δ-ferrite contents were determined: 11.7% and 10.2% in the weld metal root of the 1.29 kJ/mm and 1.55 kJ/mm weldment, respectively; 5.4% and 5.1% in the middle of the1.29 kJ/mm and 1.55 kJ/mm welded joint, respectively; and 3.2% and 2.9% in the upper part of the 1.29 kJ/mm and 1.55 kJ/mm weldment, respectively.

The coarse-grained HAZ microstructures of the 1.29 kJ/mm and 1.55 kJ/mm weldments were found to consist of bainite and martensite (Figure 8a,b), respectively. The quenched and tempered martensite of the 1.29 kJ/mm and 1.55 kJ/mm weldment base metal is shown in Figure 8c.

Figure 8 displays the coarse-grained HAZ microstructures of the weld (over-tempered) and base metal subzones of the multi-pass corner weldments made at heat inputs of 1.29 kJ/mm and 1.55 kJ/mm. The coarse-grained HAZ microstructures of both weldments considered exhibited toughness losses due to the high-temperature exposure. The microstructures were found to consist of a mixture of brittle and hard phases. The heat inputs introduced during the deposition of the subsequent passes tempered the original microstructure of the coarse-grain region, resulting in softer structures with increased toughness. However, the HAZ of the final pass remained in an untampered condition.

The microstructure of the coarse-grained HAZ region of the 1.29 kJ/mm weldment (Figure 8a) indicates the formation of smaller-volume fractions of softer constituents, i.e., lower and upper bainite. The 1.55 kJ/mm weldment (Figure 8d) consisted of a mixture of lath martensite and upper and lower bainite. The martensite to bainite (upper + lower) ratio determined was approximately 40:60. With an increase in heat input to 1.55 kJ/mm, an increasing amount of bainite was observed. However, the martensite content diminished in the microstructure of the coarse-grained region near to the fusion line. Such conditions favored the formation of bainite with a predominant amount of upper bainite in the microstructure of the 1.55 kJ/mm welded joints.

The deposition of multiple passes in high-hardness armor steel welding can result in the formation of extensively wide softened regions, especially when employing high heat inputs. Therefore, the microstructure of the over-tempered regions was subjected to close examination in the present study. The micrograph of the over-tempered regions of the multi-pass seam 1.29 kJ/mm weldment is presented in Figure 8b. The microstructure of the over-tempered zone of the1.55 kJ/mm weldment consisted predominantly of tempered martensite. An increase in heat input to 1.55 kJ/mm exerted significant effects on the microstructure of the over-tempered region of the 1.55 kJ/mm weldment (Figure 8e). The microstructure of this region consisted predominantly of tempered martensite, indicating the formation of smaller-volume fractions of softer constituents, i.e., lower and upper bainite.

The microstructure of the base metal region of both weldments considered indicates the formation of harder constituents, i.e., lath martensite (Figure 8c,f).

### 3.5. Balistic Test Results

The results of ballistic resistance testing of the welded Protac 500 joints made at a heat input of 1.55 kJ/mm are given in Table 6. The results obtained show that the initial velocities of the 7.62 × 51 mm projectile ranged from 854.896 m/s to 848.881 m/s. The equivalent shooting distance was 10 m. Two punch holes were made in this zone in the first two shootings, whereas a bulge with a protrusion was made in the third shooting. The damaged HAZ area was in the range of 70.24–90.6 mm^2^. Figure 9a,b shows the damage in the heat-affected zone of the1.55 kJ/mm weldment. The punch hole is shown in Figure 9a, whereas Figure 9b displays the protruding bulge. The damage on the inside of the heat-affected zone indicates intense plastic deformation in the direction of the projectile’s passage.

The punching area of the 1.55 kJ/mm weldment was examined using a scanning electron microscope (Figure 9b–d). The fracture surface mostly shows a smooth shear surface with sporadic areas of molten structure and ductile dimples (Figure 9b). It can be concluded that the fracture model is not uniform and that the cleavage mechanism prevails (Figure 9c,d).

The initial projectile velocities recorded in the HAZ area of the 1.29 kJ/mm weldment ranged from 850.231 m/s to 852.142 m/s (Table 6). The equivalent shooting distance was 10 m. Two plastic flows were made in this zone in the first two shootings, whereas a protruding bulge was made in the third shooting. The damaged HAZ area was in the range of 60.9–80.6 mm^2^ (Figure 10a,b). The hardness of the HAZ zone ranged from 358 HV to 521 HV. Figure 10a,b shows a plastic flow on the left side of the weld axis, whereas a bulge is discernible on the right side of the weld axis.

The initial projectile velocities recorded in base metal of the 1.29 kJ/mm weldment ranged from 849.116 m/s to 852.213 m/s. The equivalent shooting distance was 10 m. Three bulges were made in this zone in three shootings. The damaged base metal area was in the range of 30.9–40.6 mm^2^.

Figure 11 displays damage in the base metal of the 1.29 kJ/mm weldment. The hardness in this zone ranged from 499 HV to 508 HV. Figure 11 shows bulges in all three tests.

## 4. Discussion

To minimize lateral heat input in armor steel welding, which has been implicated in metal softening, it is necessary to minimize the oscillation amplitudes of the welding gun. The amplitudes of root passage, filler and right and left cover passage of 3.2 mm, 5.5 mm, 3.4 mm and 3.4 mm, respectively, were found to be the lowest amplitude values for achieving a good mixture of basic and additional material, as well as reduced heat input and HAZ softening.

The proper adjustment of preheating and inter-pass temperature requires a deeper understanding of armor steel properties relative to the continuous cooling temperature (CCT) diagram. The CCT diagram of Protac 500 reveals its quenchable nature [27]. A cooling rate of 1 °C/s produced a microstructure containing a combination of martensite and bainite, whereas a cooling rate of 7 °C/s resulted in a martensitic microstructure. Upon comparison of the microstructures of Bisaloy 500 and Protac 500 at a cooling rate of 7 °C/s, Bisaloy 500 was found to have a predominantly bainitic microstructure, whereas Protac 500 was entirely martensitic.

The microstructure formed in the HAZ is a function of the chemical composition of the steel considered and the weld thermal cycle. The main concern when employing higher heat inputs in the HHA steel welding (namely a heat input of 1.55 kJ/mm) is the formation of wide extensively softened areas in the over-tempered region that could compromise the ballistic performance of the welded structure. Conversely, the resulting prolonged cooling times temper its re-hardened HAZ areas, thus reducing the risk of HACC. These effects could compensate for employing the proposed low-level preheating in multi-pass joint welding. Figure 8 displays the microstructure of all HAZ subzones (namely the coarse-grained, fine-grained, inter-critical and over-tempered HAZ zones) of the 1.29 kJ/mm and 1.55 kJ/mm weldments.

The coarse-grained regions of all the HHA weldments were found to have mixed-mode microstructures containing martensite and upper (ferritic) and lower bainite. The effects of longer cooling times associated with increased heat input can be clearly seen in these regions. The microstructure of the weldments deposited with a higher heat input of 1.55 kJ/mm consisted of a much larger volume fraction of softer phases (upper and lower bainite).

The fine-grained regions of all the weldments considered were also found to have mixed-mode microstructures containing a mixture of martensite and upper and lower bainite. Few very fine grains of grain boundary ferrite were observed. The 1.29 kJ/mm weldment exhibited a larger number of harder phases and a higher degree of refinement compared to that of the 1.55 kJ/mm weldment. On balance, the fine-grained region of both weldments considered consisted predominantly of martensite.

An increase in heat input from 1.29 kJ/mm to 1.55 kJ/mm resulted in the formation of a wider inter-critical region. Fine-grained ferrite, granular bainite and some martensite were observed in the structure of both weldments considered.

The over-tempered subzone of the 1.29 kJ/mm to 1.55 kJ/mm weldments was found to be invariant to the heat inputs employed, consisting of tempered martensite and the original parent plate.

The microhardness HAZ values of the 1.29 kJ/mm weldment ranged from 390 HV to 523 HV, whereas the microhardness HAZ values of the 1.55 kJ/mm weldment ranged from 325 to 490 HV (Table 5). It is concluded that as heat input decreases, the hardness of the weld metal increases, which leads to ballistic protection. From the results of the Ferritescope, it can have been concluded that an increase value in the weld metal ferrite content leads to an increase in hardness [11].

Changes in the base material hardness of the 1.29 kJ/mm and 1.55 kJ/mm weldments occurred at distances of 13.5 mm and 14.2 mm from the weld axis, respectively. From a perspective of armor protection and ballistic resistance to small-arms projectiles, the selection of a heat input is important because it greatly affect the hardness of the HAZ coarse-grained area. A previous study reported that coarse-grained zone hardness values of 541 HV and 502 HV were recorded in the 0.8 kJ/mm and 1.6 kJ/mm weldments [28]. These results are similar to the results obtained in the present study. A hardness value of 523 HV was recorded in the 1.29 kJ/mm weldment (Table 5). This slightly higher hardness was achieved due to the increased hardenability of Protac 500. With a heat input of 2.37–1.33 kJ/mm, the AISI 4340 armor steel was found to have a coarse-grained zone hardness of 403–430 HV [29]. The maximum coarse-grained hardness of 443 HV was achieved with a heat input of 2.37 kJ/mm [30].

Edge preparation in the form of a double “Y” minimizes welded joint defects in the hybrid laser GMAW process [31]. A hardness value of 509 HV was recorded at 6 mm from the weld axis. With a preheating temperature of 150 °C and a heat input of 1.2 kJ/mm, the hardness values in the coarse-grained and fine-grained zone of Bisalloy 500 were 457 HV and 397 HV, respectively [32]. With the same preheating temperature and approximately the same heat input, the hardness in the fine-grained zone of Protac 500 was approximately comparable to that of Bisalloy 500. However, the course-grained zone hardness of Protac 500 was found to be significantly higher, which can be accounted for by the higher hardenability of Protac 500 (as shown in the KH diagram) [27].

The hardness results obtained show that lower heat inputs practically improve the hardness in the coarse-grained HAZ subzone. Moreover, little to no softening was observed in the over-tempered area of the optimized welds, with the hardness values exceeding the lowest hardness value of 509 HV permitted by MIL-STAN-1185 at a distance of 15.9 mm from the weld.

The results of the HAZ ballistic resistance testing for the 1.29 kJ/mm and 1.55 kJ/mm weldments, using 10 mm metal sheets, are given in Table 6. None of the three 7.62 × 51 mm projectiles fired made a punch hole in the 1.29 kJ/mm weldment. However, one of the projectiles punched through the HAZ zone of the 1.55 kJ/mm weldment. This can be explained by the diminished hardness of this zone compared to that of the 1.29 kJ/mm weldment. Using a scanning electron microscope, the dominant mechanism of cleavage was observed on the fractured surface caused by the projectile penetration. A smooth shear surface with areas of molten structure is typical of projectile-punched surfaces [18].

The criterion of no punch holes made upon three projectiles fired was only met in the case of the 1.29 kJ/mm weldment, whereas the 1.55 kJ/mm weldment failed to meet this criterion [33]. Therefore, it can be argued that a heat input of 1.29 kJ/mm provides the required degree of HAZ ballistic protection when using the 10-mm thick Protac 500 plate.

The results of ballistic resistance testing in the base material zone of the 10-mm thick Protac 500 plates considered are presented in Table 6. None of the three 7.62 × 51 mm projectiles fired made a punch hole in this zone, which can be accounted for by the optimal ductility of the zone (reflected also in the low plastic deformation sustained). Therefore, the10-mm thick Protac 500 plate was found to provide the required degree of the base material ballistic protection. The results of ballistic resistance testing in the base material zone show unequivocally that hardness is the predominant mechanical property of high- and ultra high-strength materials compared to tensile strength, yield stress and impact energy. Slight grain penetration was observed in the 1.29 kJ/mm weldment HAZ (Figure 10a,b), whereas no grain penetration was recorded in the 1.55 kJ/mm weldment HAZ (Figure 9a).

From the base material perspective (Figure 11), a hardness value exceeding 550 HV renders the material brittle, thus impairing its ballistic resistance particularly after sustaining repeated hits in the same place. A hardness of 500 HV proved optimal with regard to the ballistic resistance and toughness of the material. Slight indentations were clearly discernible on the affected surface of the material, which did not shrink due to its favorable ductility features.

The ballistic results obtained indicate that a heat input of 1.29 kJ/mm was found to be the limit for achieving the desirable armor protection and HAZ ballistic resistance of armor steels forming a 100% martensitic structure at a cooling rate of 25 °C/s. A higher heat input would impair the HAZ ballistic resistance of such steels. In the case of Protac 500, the limit for preventing grain penetration is a heat input of 1.55 kJ/mm. A heat input greater than 1.55 kJ/mm would impair the ballistic resistance of the Protac 500 HAZ.

## 5. Conclusions

The objective of this study was to examine the mechanical properties of high-hardness armor steel welded joints. Two armor steel GMAW weldments made at heat inputs of 1.29 kJ/mm and 1.55 kJ/mm were tested for ballistic protection performance. To obtain a broader insight into the welded joints considered, non-destructive testing (VT and RT) was conducted, followed by a microstructure analysis of the welded joints and measurements of their hardness and tensile strength. The effects of inter-pass and preheating temperatures were assessed by measuring the diffused and residual hydrogen in the welded joints.

On the basis of the results obtained, the following conclusions can be drawn:-The hardness of the HAZ fusion zone diminished at a heat input of 1.55 kJ/mm, resulting in the reduced ballistic protection of armored vehicles. Welded metal hardness is increased with the decrease in heat input. For impact toughness, the base metal has good toughness at any a heat input condition. However, the HAZ ballistic protection decreases notably with the heat input. The welding parameters with the 1.43 kJ/mm heat input are acceptable for high performance welded joint.-The main microstructure of WM is δ ferrite irrespective of heat input. The amount of δ ferrite in the weld metal increased with decreasing heat inputs. The microstructure in the CGHAZ changes from lath bainite/martensite to coarse granular bainite with increasing heat input.-The 1.29 kJ/mm weldment was found to exhibit significantly higher hardness levels than those of the 1.55 kJ/mm weldment.-The CCT diagram provides the starting point for assessing the allowed heat input for the proper ballistic protection of armor steels. The HAZ of armor steels forming a 100% martensitic structure at a cooling rate of 7 °C/s was found to be resistant to the 7.62 × 51 mm projectile at a heat input of 1.29 kJ/mm.-An increase in heat input leads to a ductile domain, thus reducing the ballistic performance of the 1.55 kJ/mm weldment. In the case of the 7.62 mm AP projectile, hardness and strength of the material are important for ballistic performance. Therefore, the 1.55 kJ/mm weldment was found not to be resistant to the 7.62 × 51 mm projectile penetration.

## Figures and Tables

**Figure 1 materials-14-03617-f001:**
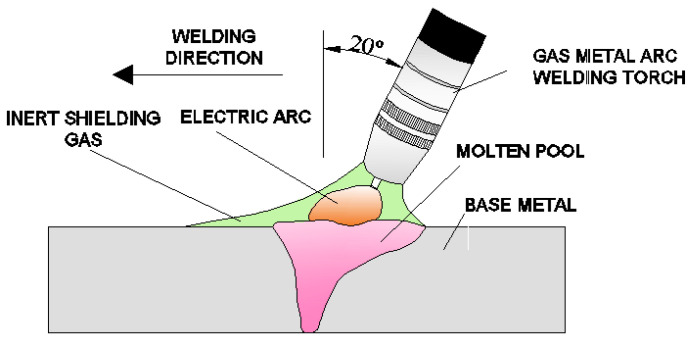
Schematic drawing of edge preparation and plate configuration.

**Figure 2 materials-14-03617-f002:**
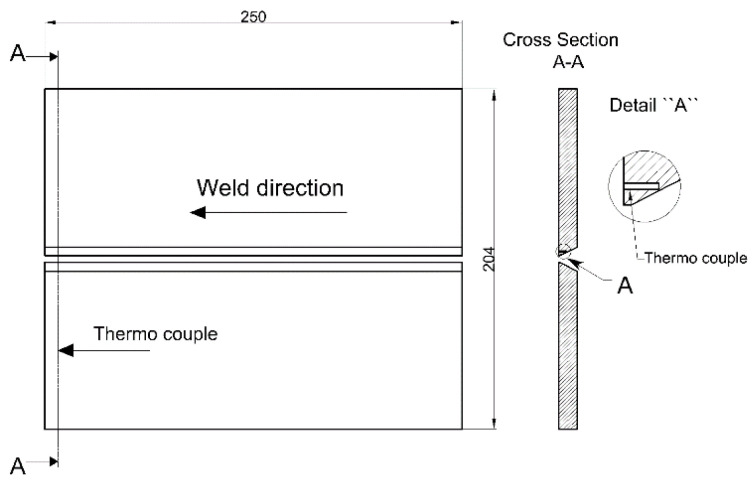
Position of thermocouple for the heat inputs considered in the Protac 500 armor steel welding.

**Figure 3 materials-14-03617-f003:**
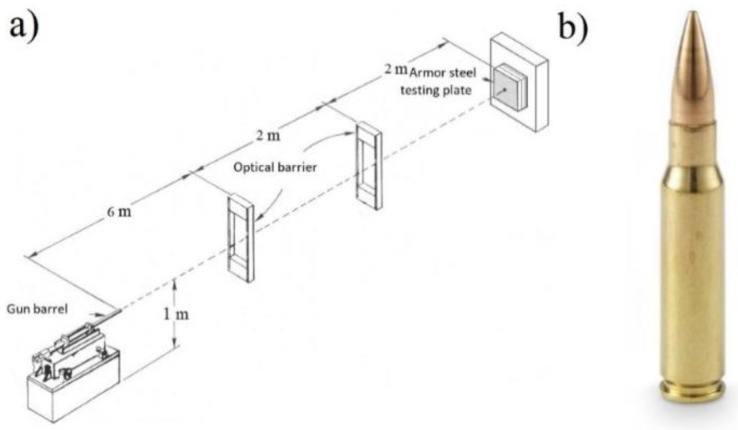
(**a**) Ballistic testing scheme; (**b**) the 7.62 × 51 mm projectile.

**Figure 4 materials-14-03617-f004:**
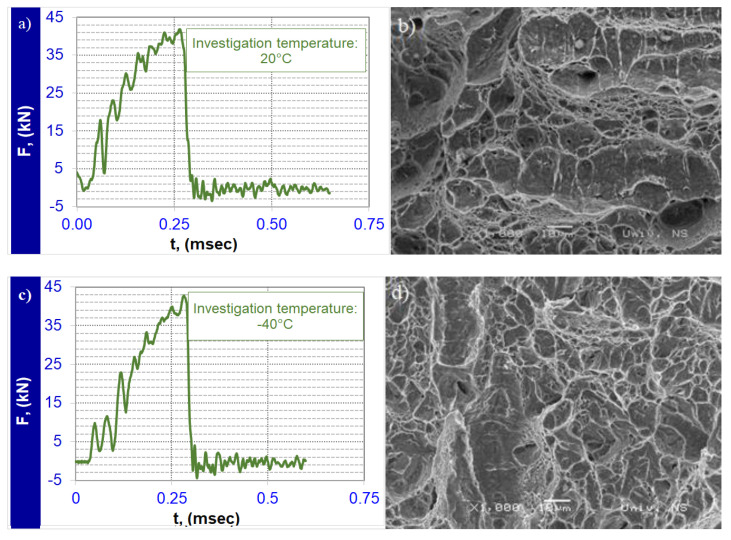
The load-time (F-t) curve recorded by oscilloscope of Charpy impact specimens fractured in base metal (**a**) at 20 °C (**c**) at −40 °C. SEM fractograph of Charpy impact specimens fractured in base metal (**b**) at 20 °C (**d**) at −40 °C.

**Figure 5 materials-14-03617-f005:**
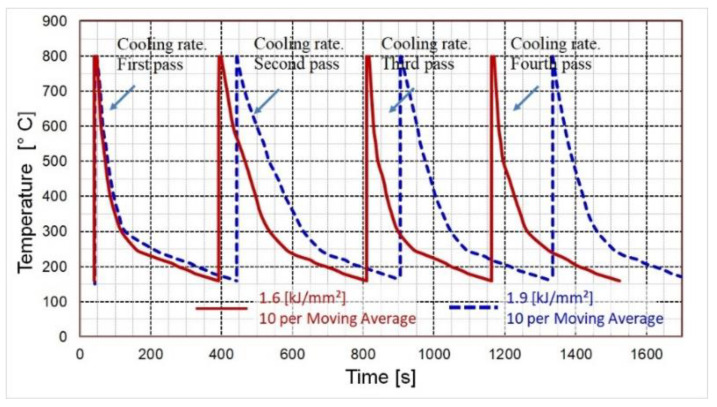
Temperature cycles for the four-pass GMAW welding process obtained by using a preheating temperature of 150 °C and an inter-pass temperature of 160 °C for heat inputs of 1.29 kJ/mm and 1.55 kJ/mm. The regression line was fitted through 10 data points.

**Figure 6 materials-14-03617-f006:**
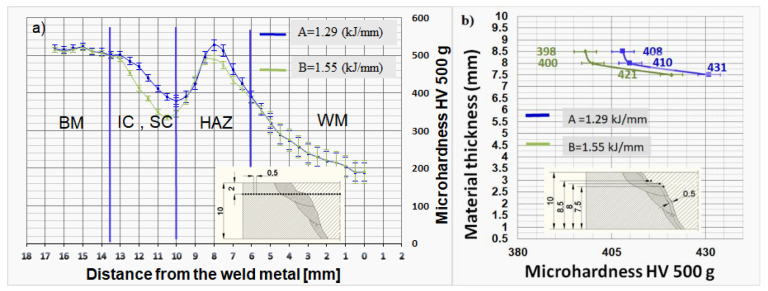
(**a**) Hardness distribution of the automated welding at heat inputs of 1.29 kJ/mm (A) and 1.55 kJ/mm (B). (**b**) Hardness distribution along the fusion line of the automated welding at heat inputs of 1.29 kJ/mm (A) and 1.55 kJ/mm (B). Each hardness value represents the mean value of three measurements performed.

**Figure 7 materials-14-03617-f007:**
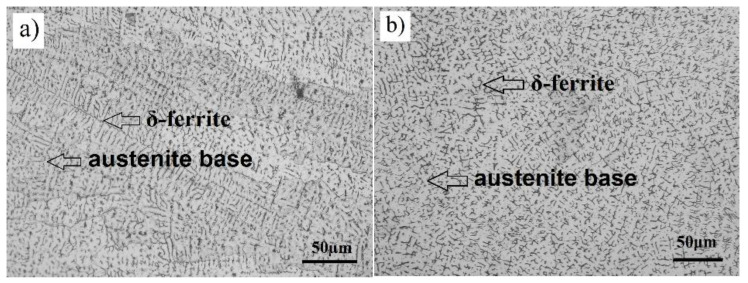
Microstructure images of the high-hardness steel weldments considered taken through an optical microscope: (**a**) the 1.55 kJ/mm weldment and (**b**) the 1.29 kJ/mm weldment.

**Figure 8 materials-14-03617-f008:**
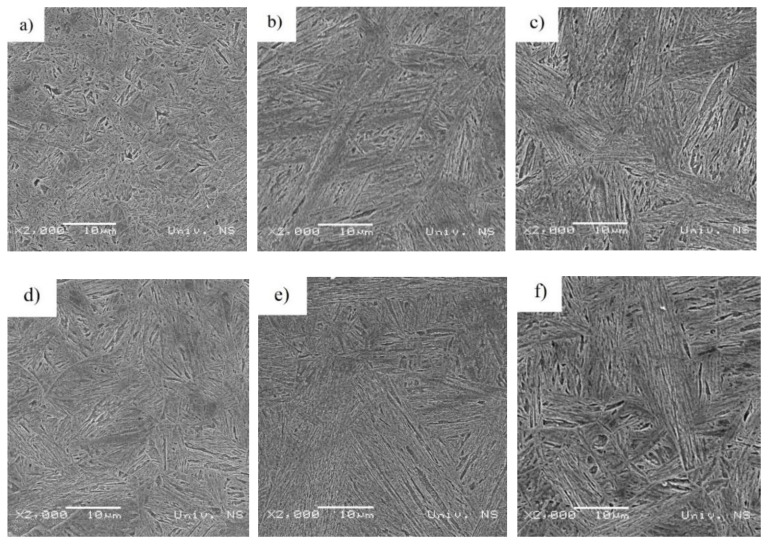
SEM microstructures of the coarse-grained HAZ (**a**) of the weld (over-tempered) (**b**) and base metal (**c**) of the 1.29 kJ/mm weldment. SEM microstructure of the coarse-grained HAZ (**d**) of the weld (over-tempered) (**e**) and base metal (**f**) of the 1.55 kJ/mm weldment.

**Figure 9 materials-14-03617-f009:**
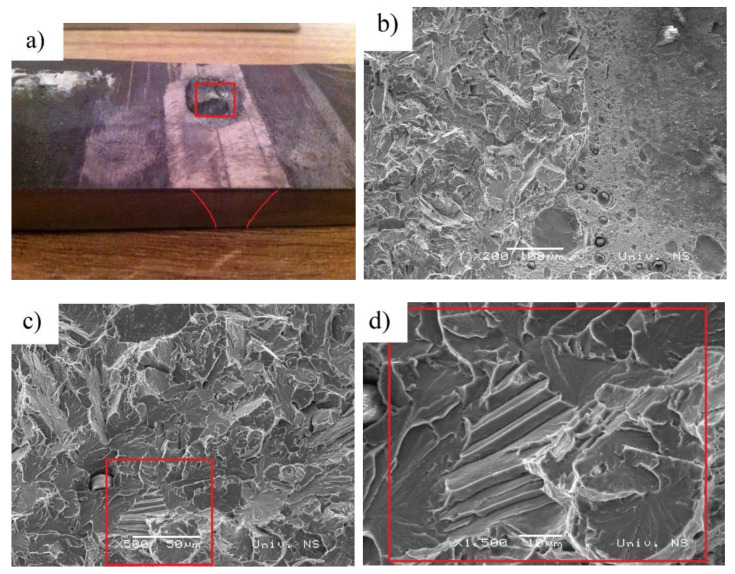
Damage in the HAZ area of the 1.55 kJ/mm weldment: (**a**) punch hole (shot 1), (**b**) molten structure and ductile dimples, (**c**) smooth shear surface and (**d**) the enlarged image of the marked (**c**) section.

**Figure 10 materials-14-03617-f010:**
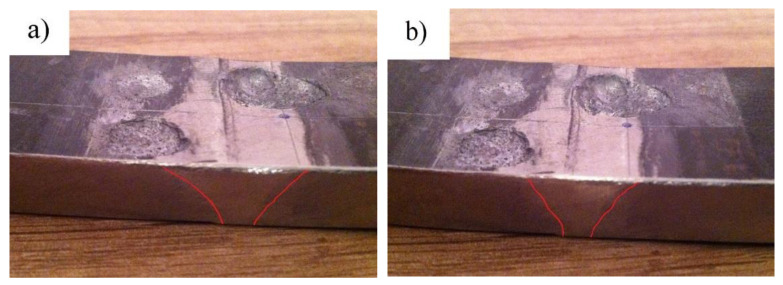
Damage in the HAZ area of the 1.55 kJ/mm weldment (**a**) left side bulge with cracks (shots 4 and 6); (**b**) right side bulge plug (shot 5).

**Figure 11 materials-14-03617-f011:**
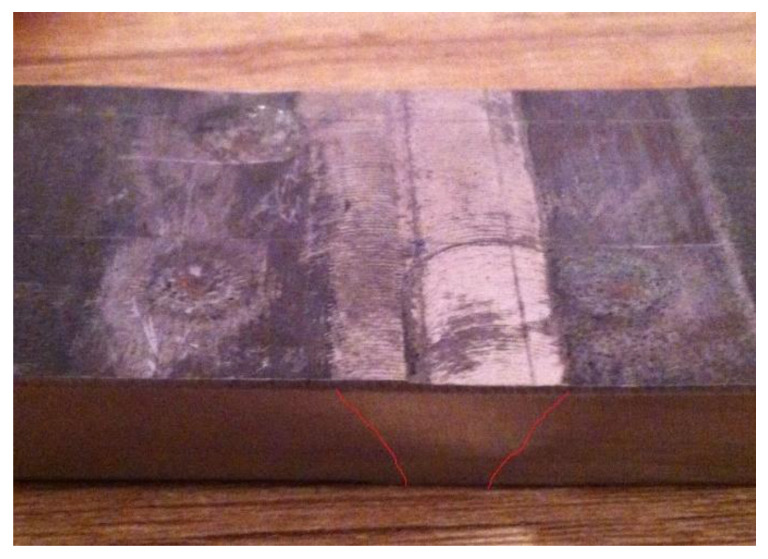
Damage in the base metal of the 1.29 kJ/mm weldment: plastic flow (shots 13, 14 and 15).

**Table 1 materials-14-03617-t001:** Chemical composition [wt. %] of the armor steel-Protac 500 and consumable ER 307.

Material	Composition [wt. %]										
	C	Si	Mn	S	Cr	P	Al	Cu	Ni	Mo	V
Protac 500	0.27	1.07	0.71	0.001	0.64	0.009	0.054	0.276	1.09	0.29	0.04
ER 307	0.08	0.89	6.29	0.001	1.76	0.014	0.01	0.08	8.24	0.13	0.03

**Table 2 materials-14-03617-t002:** Welding parameters of the Protac 500 armor steel welding.

Heat Input	Preheat Temperature	Current	Voltage	Welding Speed	Shielding Gas
	[°C]	[A]	[V]	[m/min]	Ar. + 2.5% CO_2_
1.29	160	193	25	0.18	
1.55	160	215	25.5	0.17	

**Table 3 materials-14-03617-t003:** Instrumented Charpy impact energy properties of base metal in the range of 20 °C to −40 °C.

Test Temperature	Total Absorbed Energy (E_T_)	Fracture Initiation Energy (E_I_)	Fracture Propagation Energy (E_p_)
[°C]	[J]	[J]	[J]
20	30.4	4.8	35.2
0	27.6	4.5	32
−20	25.9	3.9	29.8
−40	24.9	2.9	27.7

**Table 4 materials-14-03617-t004:** Cooling times of the four-pass 1.29 kJ/mm and 1.55 kJ/mm GMAW processes.

Heat Input	Heat Input 1.29 [kJ/mm]	Heat Input 1.55 [kJ/mm]
Cooling time from the first pass to the peak temperature of the last pass [s]	1122	1293
Cooling time from the first pass to the 160 °C temperature of the last pass [s]	1484	1696

**Table 5 materials-14-03617-t005:** Micro hardness of the 1.29 kJ/mm and 1.55 kJ/mm weldments.

Heat Input	Maximum Hardness/Distance from the Weld Centerline			
	Weld Metal	HAZ Maximum Value	HAZ Minimum Value	Base Metal
[kJ/mm]	[HV]/[mm]	[HV]/[mm]	[HV]/[mm]	[HV]/[mm]
1.29	290 ± 8/4.5	523 ± 8/8.0	390 ± 8/10.0	502 ± 17/14.0
1.55	290 ± 7/4.5	490 ± 7/8.5	325 ± 7/10.5	503 ± 15/14.0

**Table 6 materials-14-03617-t006:** Results of ballistic resistance testing of the welded Protac 500 joints made at a heat inputs of 1.29 and 1.55 kJ/mm.

Serial Number	Heat Input	Position	Initial Speed V10	Equivalent Shooting Distance	Angle of Impact Relative to the Projectile Trajectory	Type of Damage
	[kJ/mm]		[m/s]	[m]	[°]	
4	1.55	HAZ	852.142	10	90	punch hole
5	1.55	HAZ	851.321	10	90	punch hole
6	1.55	HAZ	850.231	10	90	bulge
7	1.55	Base metal	849.116	10	90	plastic flow
8	1.55	Base metal	850.212	10	90	plastic flow
9	1.55	Base metal	852.313	10	90	plastic flow
10	1.29	HAZ	852.048	10	90	bulge
11	1.29	HAZ	851.254	10	90	bulge
12	1.29	HAZ	850.358	10	90	bulge
13	1.29	Base metal	849.742	10	90	plastic flow
14	1.29	Base metal	850.343	10	90	plastic flow
15	1.29	Base metal	852.259	10	90	plastic flow

## Data Availability

Data is contained within the article.

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
