# Peer review of "Effect of Heat Input on the Ballistic Performance of Armor Steel Weldments"

_materials, 2021, doi:10.3390/ma14133617_

Round 1

Reviewer 1 Report

  1. The current study evaluates the influence of temperature on the performance of armoured steel subjected to ballistic impact loadings. For this, the authors study the impactor perforation and the resistance of the armour to perforation. The authors use different heat energies and analyse the heat affected zones.
  2. Please consider reviewing the abstract and highlight the novelty, major findings and conclusions.
  3. The introduction requires significant improvements, there is very little literature on the subject of the manuscript. The authors must provide comprehensive literature review, summarizing past studies, report on what they did and what were their main findings then explain how does the current study brings new knowledge and difference to the field.
  4. After line 50 the authors must answer the following question: What is the research gap did you find from the previous researchers in your field? Mention it properly. It will improve the strength of the article.
  5. The authors should add details of the impactor and the armoured steel material in a table and also add the mechanical and thermal properties of the armoured steel.
  6. Line 68-73 add the chemical composition data in a table and reference it if needed
  7. Why the authors choose those specific parameters in Table 1? Are they based on literature or real life in service conditions or recommendations or just randomly chosen for this study?
  8. Change the title of section 3 to Results and discussion
  9. In Table 3 there is only one odd data between the two showing 390 and 325 HV, all the others can be said to be consistent? Please elaborate on this further
  10. Table 3 please add the locations of the hardness measurement
  11. Figure 5 please make the sizes of the two graphs the same
  12. From Figure 5 a it does not seem to be much influence of the heat on change of hardness? Please elaborate on this issue further and support with references
  13. Figure 6 add some arrows and text on the SEM images to tell the readers what are they looking at in them
  14. Consider combining figures 6 and 7 into one figure
  15. Consider combining figures 8, 9 and 10
  16. Combine tables 4 and 5 in one table (recommended) most of the columns are the same just the type of damage is different!
  17. In the discussion section, it seems that the authors are adding references to some sentences however this does not make any sense? For example line 361 the authors write a sentence and then add a reference at the end. However this sentence does not seem to need a reference? This is not acceptable, the authors should check this issue everywhere in the discussion section and provide more appropriate and in depth discussion and critical analysis.
  18. The results are merely described and is limited to comparing the experimental observation. The authors are encouraged to include a more detailed discussion section and critically discuss the observations from this investigation with existing literature.

Author Response

1. Please consider reviewing the abstract and highlight the novelty, major findings and conclusions.

Respond: Abstract was extended with the sentence of major findings. “The result showed that the ballistic resistance of heat affected zone exist as the heat input was decreased on 1.29 kJ/mm. It was found that 1.55 kJ/mm not have ballistic resistance.”

2. The introduction requires significant improvements, there is very little literature on the subject of the manuscript. The authors must provide comprehensive literature review, summarizing past studies, report on what they did and what were their main findings then explain how does the current study brings new knowledge and difference to the field.

Respond: the introduction was extended with three paragraphs.

3. After line 50 the authors must answer the following question: What is the research gap did you find from the previous researchers in your field? Mention it properly. It will improve the strength of the article.

Respond: Authors added sentence According to this statements, Protac 500 require increase preheat and inter pass temperatures.

4. The authors should add details of the impactor and the armoured steel material in a table and also add the mechanical and thermal properties of the armoured steel.

Respond: In phd research I have investigated instrumented Charpy impact test in all weld metal zones. Investigation in base metal I have not published anywhere. I have added this investigation in research paper.

5. Line 68-73 add the chemical composition data in a table and reference it if needed

Respond: Table has been added.

6. Why the authors choose those specific parameters in Table 1? Are they based on literature or real life in service conditions or recommendations or just randomly chosen for this study?

Respond: Parameters are chosen based on investigation of previously researcher. But the parameters, from prevously research, are to wide and not precise. 

7. Change the title of section 3 to Results and discussion

Respond: Section is renamed to Results and discussion.

9. Table 3 please add the locations of the hardness measurement

Respond: “The hardness of the Protac 500 welded joints was herein tested 2 mm under the upper welding surface at heat inputs of 1.29 kJ/mm and 1.55 kJ/mm. The hardness of both heat input samples considered was measured along the fusion line for achieving optimum hardness in this critical zone and along the edge of the weld metal.”

10. Figure 5 please make the sizes of the two graphs the same

Respond: It has been corrected.

11. Combine tables 4 and 5 in one table (recommended) most of the columns are the same just the type of damage is different!

Respond: It has been corrected.

12. From Figure 5 a it does not seem to be much influence of the heat on change of hardness? Please elaborate on this issue further and support with references

Respond: following sentences are added in discussion section. It is concluded that as heat input decreases, the hardness of the weld metal increases, which leads to ballistic protection. From the results of the Ferritescope, it can have been concluded that an increase value in the weld metal ferrite content leads to an increase in hardness [11].

13. Figure 6 add some arrows and text on the SEM images to tell the readers what are they looking at in them

Respond: Arrows are add.

14. Consider combining figures 6 and 7 into one figure

Respond: I try but it is not look unify in columns. On Figure 6 are two pictures, and on Figure 7 are three picture in columns.

15. Consider combining figures 8, 9 and 10

Respond: if combine figures into the one, we will have too many sentences in Figure explanation. Explanation is bellow given

Figure 9. Damage in the HAZ area of the 1.55 kJ/mm weldment: a) punch hole (shot 1), b) molten structure and ductile dimples, c) smooth shear surface, and d) the enlarged image of the marked Fig. 9 c section. Damage in the HAZ area of the 1.55 kJ/mm weldment a) left side bulge with cracks (shots 4 and 6); b) right side bulge plug (shot 5). Damage in the base metal of the 1.29 kJ/mm weldment: plastic flow (shots 13, 14 and 15).

16. Combine tables 4 and 5 in one table (recommended) most of the columns are the same just the type of damage is different!

Respond: It has been corrected. Two tables are combined into the one. Thank you for recommendation.

17. In the discussion section, it seems that the authors are adding references to some sentences however this does not make any sense? For example line 361 the authors write a sentence and then add a reference at the end. However this sentence does not seem to need a reference? This is not acceptable, the authors should check this issue everywhere in the discussion section and provide more appropriate and in depth discussion and critical analysis.

Respond: I check this issue everywhere and provide more appropriate discussion.

18.The results are merely described and is limited to comparing the experimental observation. The authors are encouraged to include a more detailed discussion section and critically discuss the observations from this investigation with existing literature.

Respond: Conclusion  is changed, it include a more detailed explanation.

The hardness of the HAZ fusion zone diminished at a heat input of 1.55 kJ/mm, resulting in the reduced ballistic protection of armored vehicles. Welded metal hardness are increased with the decrease in heat input. For impact toughness, the base metal have good toughness at any a heat input condition. However, the HAZ ballistic protection decreases notably with the heat input. The welding parameters with the 1.43 kJ/mm heat input are acceptable for high performance welded joint.

The main microstructure of WM is δ ferrite irrespective of heat input. The amount of δ ferrite in the weld metal increased with decreasing heat inputs. The microstructure in the CGHAZ changes from lath bainite/martensite to coarse granular bainite with increasing heat input.

Reviewer 2 Report

Dear Authors,

I have reviewed paper "Effect of heat input on the ballistic performance of armor steel weldments".

It fulfills the aims and scope of Materials. Investigations are novel and important. After some improvementes, the paper could be considered for publishing.

My biggest objection concerns the presented values of heat input. The calculations are not the same, as presented in the text. The equation is. The "k" factor is different for different welding processes.

My suggestions and comments are presented below.

General remarks:

  • You have presented 30 references. Only one has been published since 2019. In my opinion, you should extend the list of references by newly published papers. It provide to increasing the visibility of your paper in scientific databases.
  • The style of your paper is away from template in many places, e.g., the style of references is much different. Please check and improve this issue.
  • There is no information about welding process in the abstract. Which process was used in your investigations?

Introduction:

  • This section is quite short. You should extgend it.
  • I propose to describe the usage of armor steel joints more wider. Now, only in line 26 is one information in this field. Please show other exampels.
  • Moreover, you should describe the welding processes, which are used for joining the investigated group of steel. Their advantages and disadvantages should be presented. It will underline the necessity of your investigations and their novelty.
  • Line 26 - should be [1-3] not [1,2,3]. The same in the rest of your text.
  • The same, as in the abstract. The information about used welding process should be stated near the novelty.

Material and Experiments:

  • Firstly, I propose to change "Material" to "Materials", because you described the base material and filler material = you described materials.
  • Lines 60-62 - this is obvious. You should remove this lines.
  • The description of used materials is poor. You have show the chemical composition and mechanical properties of base and filler material. The properties will prove, that you have chosen proper filler material.
  • Line 72 " The chemical composition of the welded joints was determined using theARL 722460 spectrometer." In this section, you have not presented any composition of welded joint. Only the composition of materials was presented.
  • Table 1 - how the heat input values were calculated? There is no information about used k (efficiency factor). Moreover, following the equation of for ql calculations (with relevant k factor), the values presented in the table, do not allow to get presented heat input values.
    In the first issue the calculated heat input is 1.287 kJ/mm, not 1.6 kJ/mm. In the second - 1.55, not 1.9 kJ/mm.

Results:

  • That does it mean "Δt6/2"? Moreover, in the welding engineering, the crucial time is time t8/5. Why you have not presented and discussed this time?
  • The quality of Fig. 5 is poor. Moreover pictures a and b ar echaracterized by much different size.
  • There is no information about etching during microscopic testing.
  • Line 209 - what does it mean "in the upper part"? Tests were carried out neat the axis of the weld, near the fusion line?
  • Fig. 7 - Pictures were takken near the face of the weld, near the groove? It is crucial, becaluse the heat from stitches play significant role in microstructures of lower layers.

Discussion and Conclusions:

  • This sections is clear.

However, please comment the presented values of heat input. In my opinion, these values are wrong.

Author Response

Dear Reviewer

My biggest objection concerns the presented values of heat input. The calculations are not the same, as presented in the text. The equation is. The "k" factor is different for different welding processes.
Respond:

Dear reviewer thank you very much, for notice such big mistake. I was in the focus of the welding parameters to get good welding joint, without cracks and pores where projectile will not punch heat affected zone.

Thank you, even for such big mistake, you consider my work to be publish. I have added k factor into calculation.

My suggestions and comments are presented below.

Thank you very much for the revision professor. I

General remarks:

  • You have presented 30 references. Only one has been published since 2019. In my opinion, you should extend the list of references by newly published papers. It provide to increasing the visibility of your paper in scientific databases.

Respond:

Dear reviewer I have changed references with the sciences papers published from 2018 till 2021.

  • The style of your paper is away from template in many places, e.g., the style of references is much different. Please check and improve this issue.

Respond:

I did not know that my paper could be consider for publication in Materials. I sent my research work in order to see is it possible to be consider for publish in your journal. I am working in automotive industry, with responsibility for turbocharger material, and I was I doubt whether my paper can be considering for publishing. The style is changed now according to template.

  • There is no information about welding process in the abstract. Which process was used in your investigations?

Respond:

Dear reviewer I have add the one sentence in abstract, where I informed which welding process is used.

Introduction:

  • This section is quite short. You should extend it.

Respond:

Introduction section is extended with welding information.

  • I propose to describe the usage of armor steel joints more wider. Now, only in line 26 is one information in this field. Please show other examples.

Respond:

Thank you for this remark. In the research paper I add the one paragraph with welding information.

  • Moreover, you should describe the welding processes, which are used for joining the investigated group of steel. Their advantages and disadvantages should be presented. It will underline the necessity of your investigations and their novelty.

Respond:

I have extended this information.

  • Line 26 - should be [1-3] not [1,2,3]. The same in the rest of your text.

Respond:

It is changed according to Materials template.

  • The same, as in the abstract. The information about used welding process should be stated near the novelty.

Material and Experiments:

  • Firstly, I propose to change "Material" to "Materials", because you described the base material and filler material = you described materials.

Respond:

I agree, we have different materials. Thank you. Instead of Material I change to be Materials.

  • Lines 60-62 - this is obvious. You should remove this lines.

Respond:

Lines 60-60 are removed.

  • The description of used materials is poor. You have show the chemical composition and mechanical properties of base and filler material. The properties will prove, that you have chosen proper filler material.
  • Line 72 " The chemical composition of the welded joints was determined using theARL 722460 spectrometer." In this section, you have not presented any composition of welded joint. Only the composition of materials was presented.

Respond:

I did not understand this remark. I analyzed chemical composition of base metal and chemical composition weld metal, extracted from the weld joint.

  • Table 1 - how the heat input values were calculated? There is no information about used k (efficiency factor). Moreover, following the equation of for ql calculations (with relevant k factor), the values presented in the table, do not allow to get presented heat input values.
    In the first issue the calculated heat input is 1.287 kJ/mm, not 1.6 kJ/mm. In the second - 1.55, not 1.9 kJ/mm.

Respond:

Thank you for notice such big mistake. Even such mistake you are consider this paper to be published. Off course, I add K factor, 0.8, for GMAW process. The new heat input, 1.29 and 1.55 kJ/mm are replaced in research paper.

Results:

  • That does it mean "Δt6/2"? Moreover, in the welding engineering, the crucial time is time t8/5. Why you have not presented and discussed this time?

Respond:

I add paragraph with explanation why we need use temperature from 600 to 200 °C.

  • The quality of Fig. 5 is poor. Moreover pictures a and b ar echaracterized by much different size.

Respond:

Regarding this issue, characters are unified as much as possible.

  • There is no information about etching during microscopic testing.

Respond:

I add sentence about etching.

  • Line 209 - what does it mean "in the upper part"? Tests were carried out neat the axis of the weld, near the fusion line?

Respond:

Amount of δ - ferrite is not the same in root of the weld metal zone c), Figure1, or in the middle of the weldment b) or at upper position a). the maximum value is in the root position c), then at middle b) and least at upper position a).

Fig. 7 - Pictures were takken near the face of the weld, near the groove? It is crucial, becaluse the heat from stitches play significant role in microstructures of lower layers.

Discussion and Conclusions:

  • This sections is clear.

However, please comment the presented values of heat input. In my opinion, these values are wrong.

Round 2

Reviewer 1 Report

All questions were answered paper can be accepted

Reviewer 2 Report

Dear Authors,

Thank you for your response. Your efforts are appritiate, paper has been improved a lot. Moreover, you hav commented all my remarks. Accordingly these reasons, I recommand your work for publishing.

Best regards,

Reviewer